# Mechanisms Supporting the Use of Beta-Blockers for the Management of Breast Cancer Bone Metastasis

**DOI:** 10.3390/cancers13122887

**Published:** 2021-06-09

**Authors:** Maria-Bernadette Madel, Florent Elefteriou

**Affiliations:** 1Department of Orthopedic Surgery, Baylor College of Medicine, Houston, TX 77030, USA; Maria-Bernadette.Madel@bcm.edu; 2Department of Molecular and Human Genetics, Baylor College of Medicine, Houston, TX 77030, USA

**Keywords:** sympathetic nervous system, bone, osteoblasts, norepinephrine, adrenergic receptors, breast cancer, tumor microenvironment, metastasis

## Abstract

**Simple Summary:**

Bone represents the most common site of metastasis for breast cancer and the establishment and growth of metastatic cancer cells within the skeleton significantly reduces the quality of life of patients and their survival. The interplay between sympathetic nerves and bone cells, and its influence on the process of breast cancer bone metastasis is increasingly being recognized. Several mechanisms, all dependent on β-adrenergic receptor signaling in stromal bone cells, were shown to promote the establishment of disseminated cancer cells into the skeleton. This review provides a summary of these mechanisms in support of the therapeutic potential of β-blockers for the early management of breast cancer metastasis.

**Abstract:**

The skeleton is heavily innervated by sympathetic nerves and represents a common site for breast cancer metastases, the latter being the main cause of morbidity and mortality in breast cancer patients. Progression and recurrence of breast cancer, as well as decreased overall survival in breast cancer patients, are associated with chronic stress, a condition known to stimulate sympathetic nerve outflow. Preclinical studies have demonstrated that sympathetic stimulation of β-adrenergic receptors in osteoblasts increases bone vascular density, adhesion of metastatic cancer cells to blood vessels, and their colonization of the bone microenvironment, whereas β-blockade prevented these events in mice with high endogenous sympathetic activity. These findings in preclinical models, along with clinical data from breast cancer patients receiving β-blockers, support the pathophysiological role of excess sympathetic nervous system activity in the formation of bone metastases, and the potential of commonly used, safe, and low-cost β-blockers as adjuvant therapy to improve the prognosis of bone metastases.

## 1. Introduction

Metastasis to distant organs represents a common and fatal complication in breast cancer patients [1,2]. About 70% of these patients experience metastatic bone disease, which can occur years after completion of the treatment of the primary tumor. Associated with a 5-year overall survival rate of approximately 23%, bone metastases represent a particularly unfavorable prognosis and can only be treated palliatively [1,3,4,5].

Cancer cell metastasis is a multistep process characterized by five major interrelated pathological events. As a consequence of their uncontrolled growth and low oxygen tension, tumor cells secrete proangiogenic factors that contribute to tumor vascularization. The high vascularity of the primary tumors provides a chance for metastatic cells to disseminate to other organs following detachment from the primary tumor, infiltration into the surrounding stroma, and migration through the basement membrane supporting the endothelium of local blood and/or lymphatic vessels. Intravasation into the blood/lymphatic circulation is followed by hematogenous or lymphagenous dissemination of a minimal number of anoikis-resistant tumor cells to distant organs. In these tissues, cancer cells extravasate into the tissue stroma in response to endothelial and stroma factors to form microfoci that can remain dormant or grow into full-blown metastatic macrofoci. Those macrofoci formed in skeletal tissues will impact skeletal homeostasis, structure and endocrine function [6,7,8,9,10]. These steps are driven in the primary and metastatic tumor microenvironment by direct cell-to-cell and paracrine interactions between cancer cells and different stromal cells [8,11,12,13,14,15,16]. 

The skeleton represents a preferred organ for metastasis of breast, prostate, and lung cancers [17,18,19,20,21,22]. It is a large, richly vascularized organ, abundantly innervated by sensory nerves found in the cancellous and mainly periosteal bone compartment, as well as sympathetic nerves, mostly associated with the bone marrow vasculature, sprouting toward bone surfaces [23,24,25,26,27,28,29,30,31,32]. Recent studies in mice have shown that sensory nerves, in addition to mediating pain postnatally, play an important role in the early formation of the skeleton and in the response of the skeleton to mechanical loading and fracture [33,34]. In contrast, sympathetic nerves were shown to be involved in the process of bone remodeling in adult mice, under the control of hypothalamic and brainstem centers in the central nervous system, with likely relevance to age-related bone loss, a condition concurrent with the time of most skeletal metastatic occurrences.

Over the past two decades, the relevance of skeletal innervation by sympathetic nerves extended to the process of bone metastasis, starting with the observation that chronic stress was associated with enhanced recurrence rates in breast cancer patients, and with metastasis formation, reduced overall survival, and poor patient prognosis [35,36,37,38,39,40,41,42,43,44], whereas β-blocker use was associated with increased relapse-free survival [40,42,44,45,46,47,48,49]. Chronic stress stimulates activity of the hypothalamic–pituitary axis (HPA) and the sympathetic nervous system (SNS), resulting in the release of norepinephrine (NE) from sympathetic nerve terminals and the stimulation of post-synaptic β-adrenergic receptors (βARs) on target cells, including bone [50] and cancer cells [51,52,53,54,55,56,57] (reviewed in [58]). These clinical observations were the first to support a putative role of excess sympathetic outflow in the mechanisms leading to the successful establishment of disseminated cancer cells into the skeleton. Since then, studies in vitro and in preclinical mouse models have refined our understanding of the mechanisms whereby sympathetic nerves can impact the metastatic process, and have revealed the complexity of these mechanisms. The effect of sympathetic nerves on the early stages of metastasis has recently been reviewed [58]. In the present review, we focus more specifically on mechanisms supporting the role of the SNS in the establishment of breast cancer cell metastases in the skeleton, and on how this knowledge supports the potential of βAR blockade as a new therapeutic strategy for a better management of patients with skeletal breast cancer metastases.

## 2. Sympathetic Innervation of the Skeleton and Evidence for an Interplay with the Process of Bone Metastasis

Stress has well-defined repercussions on body homeostasis. In the short-term, it stimulates the HPA and the SNS, resulting in the release of glucocorticoids and epinephrine from the adrenal glands and of NE from sympathetic nerve terminals to support the typical fight-or-flight response and survival. Chronic stress, however, has an overall negative impact on the body and has been associated with multiple pathological conditions, including coronary heart disease [59,60], obesity [61,62], and gastrointestinal diseases [63,64,65,66,67]. In the context of bone health, chronic psychological stress and severe depression have been associated with higher fracture risk and low bone mineral density (BMD) [68,69,70,71,72,73,74,75,76,77,78,79]. Although this association is likely multifactorial, one of its components is thought to be the abundant innervation of the skeleton by sympathetic nerves and the action of these nerves on bone cells and bone remodeling. Sympathetic nerve fibers are indeed located in close vicinity to bone cells, including osteoblasts, osteoclasts, and bone marrow cells, which all express post-synaptic βARs [51,52,53,54,55,56,57,80,81,82]. Studies in mice, using isoproterenol (ISO) as a surrogate for endogenous nerve-released NE, showed that βAR agonists cause bone loss by inhibiting osteoblast proliferation and by promoting their production of pro-osteoclastogenic factors and bone resorption [83,84]. On the other hand, mice lacking β2AR globally or specifically in osteoblasts were characterized by a high bone mass, and similar outcomes were observed in mice administered with the β-blocker propranolol [83,84,85]. These studies thus supported the critical role of βAR signaling in bone remodeling and pointed to osteoblasts as one of the critical targets of sympathetic nerves for their action on bone homeostasis.

β-blockers are commonly used to treat cardiovascular conditions, and retrospective studies have provided additional clinical insights to support the role of sympathetic nerves in the process of bone remodeling, while confirming the clinical relevance of the aforementioned preclinical findings [86,87,88,89,90,91,92,93,94]. These studies also revealed that β-blocker use at time of diagnosis in patients with triple negative breast cancer (but also non-small cell lung cancer, hemangiomas, and ovarian cancer) was associated with prolonged disease-free survival and reduced metastasis development and tumor recurrence [45,46,47,48,49,95,96,97,98,99,100,101,102]. Since no correlation was found between the post-diagnostic use of β-blockers and breast cancer progression [103], these observations implied that the βAR-dependent mechanisms leading to recurrence might mainly affect early stages of metastatic dissemination. As 70% of breast cancer patients succumb with skeletal metastases, these observations also suggested that sympathetic nerves, besides regulating bone homeostasis, participate in the molecular events that stimulate skeletal metastasis. 

There are multiple mechanisms by which stress and sympathetic nerve activation can contribute to tumor metastasis. These include a direct effect on tumor cells, an indirect effect on the stroma of the primary tumor or distant metastatic tumor, and/or an effect on immune surveillance. Additional mechanisms might be involved, and they are not mutually exclusive. For instance, βAR stimulation in tumor cells and surrounding stromal elements in the primary tumor, such as tumor-associated macrophages and vascular endothelial cells, promotes molecular processes involved in tumor progression. These include DNA damage repair mechanisms by activation of the ataxia–telangiectasia and Rad3-related/p21 pathway [104,105,106], oncogene activation including Src [107], HER2 signaling [108,109], or resistance to chemotherapy-mediated cell death [110,111,112,113]. In line with the clinical findings associating stress and enhanced recurrence rate in cancer patients [42,43,44,45,46,47,48,49], in vitro studies demonstrated that NE decreases the efficacy of chemotherapeutics [104], and pharmacological βAR activation in breast cancer cells was shown to promote tumor cell growth and migration [114], as well as the expression of inflammatory and chemotactic cytokines required for metastasis and carcinogenesis [53,115,116,117,118]. These effects were abolished when using the β-blocker propranolol [53,115,116,117]. On the other hand, other studies showed an anti-proliferative effect of βAR agonists [118,119,120]. Results regarding a direct effect of catecholamines or βAR agonists on cancer cells are thus conflicting, but this is to be expected based on the in vitro nature of these assays, the focus on unique cell lines with specific characteristics, and the different doses of drugs used. The expression of αARs and βARs also differs between cancer cells lines and across molecular subtype (ER, PR, and HER2 status, reviewed in [58]). β2AR is the most widely expressed βAR in basal and luminal breast cancer cell lines and in tumor samples from patients with breast cancer [58,121,122,123]. β2AR expression was found to correlate with poor prognosis of ER^−^ breast cancer patients [123]. In another study, β2AR expression was associated with lower disease-free survival and higher lymph node metastasis rates in a small cohort of HER2^+^ breast cancer patients [121], but opposing results were obtained in another small cohort of HER2^+^ patients where it was associated with improved disease-free survival [124]. Inconsistencies regarding the type or level of expression of adrenergic receptors and the effect of their agonists in breast cancer cells thus limit confidence that modulation of βAR signaling in cancer cells may be of any solid predictive and therapeutic value, although this needs to be further examined.

β2AR is also widely expressed in host stromal tissues and βAR agonists can have distinct cellular targets and effects in different tissues and at specific stages of the metastatic process. The effect of βAR agonists and antagonists on this process requires all findings to be interpreted with this in mind. Further adding complexity to the mechanisms whereby sympathetic nerves influence breast cancer tumor progression is the difference between pharmacological βAR agonists and endogenous NE released by nerves in terms of adrenergic signaling and action. Pharmacological βAR agonists indeed acutely stimulate post-synaptic receptors on cancer cells or host cells, in contrast to endogenous nerve-released NE following SNS activation via chronic stress or other means, which is buffered by homeostatic mechanisms aimed at controlling excess SNS activation. Among the multiple components of this homeostatic system, the norepinephrine transporter (NET), located in the plasma membrane of noradrenergic neurons, is responsible for NE re-uptake and recycling. It serves as the primary mechanism for inactivating noradrenergic signaling and terminating the short-term biological effects of NE in the synaptic cleft [125,126], regulating adrenergic neurotransmission in the brain and peripheral organs [127,128]. NET is not only expressed in presynaptic neurons but also in bone cells, including mature osteoblasts and osteocytes [129,130] and its expression decreases during aging in mice [130], pointing to a potential role of this transporter in the regulation of bone remodeling, and possibly bone metastasis, and potentially explaining conflicting results between the effect of pharmacological βAR agonists and endogenous SNS activation on this process. 

Although the assessment of an immune component is excluded in most studies related to cancer bone metastasis because of the common use of immunocompromised mice that do not reject human cancer cells, it is known that most immune cells express βARs and that SNS activation plays an integral role in the regulation of the anti-tumor immune response. For instance, sympathetic activation can promote the development of myeloid-derived suppressor cells (MDSCs) to favor an immunosuppressive environment [131] as well as macrophage tumor infiltration [132,133]. βAR signaling stimulates the expression of TGFβ, VEGF, IL-6, MMP9, and PTGS2 by macrophages, thereby promoting tumor progression and inhibiting the transcription of type I and II interferons [132,134,135], which are important in cell-mediated immune responses against cancer. Furthermore, βAR signaling can suppress the cytotoxic function of T cells and NK cells and contribute to the dissemination of cancer cells [136,137,138,139]. However, in vivo preclinical studies have demonstrated that stimulation of the SNS favors tumor metastasis in the absence of NK and cytotoxic T cells [53,110,132,140,141], thus suggesting the existence of T- and NK-cell-independent mechanisms contributing to the engraftment of metastatic tumor cells into the skeleton. 

## 3. SNS-Induced Bone Stromal-Dependent Mechanisms Promote Skeletal Colonization by Cancer Cells

Multiple observations support the notion that sympathetic nerves act on cells of the osteoblast lineage to promote the formation of skeletal metastases. An important initial suggestive evidence of such mechanisms was the observation that hematopoietic stem cells from the bone marrow environment were mobilized into the circulatory system by sympathetic nerve signals and the induction of stromal-cell-derived factor 1 (SDF1/CXCL12), a chemokine also involved in the dissemination of metastatic cells into the skeleton [142,143]. Osteoblasts, osteocytes, and osteoclasts predominantly express β2AR and can thus respond to NE released by sympathetic nerves [51,52,83,84,144,145,146]. Angiogenesis represents a necessary component of both invasive tumor growth and distant metastasis. One of the main inducers of angiogenesis is vascular endothelial growth factor (VEGF) [147,148,149,150], whose expression correlates with increased primary tumor microvasculature and malignancy and metastasis in breast cancer patients [151]. In vitro studies revealed that VEGF synthesis is increased upon βAR stimulation of tumor cells [81,152] and animal studies showed that high endogenous sympathetic outflow increases VEGF and vascular density in primary tumors [53,153,154]. Besides its direct action on the primary tumor, the SNS also influences the vascular density of the bone marrow microenvironment. The skeleton is a highly vascularized tissue, and chondrocytes and osteoblasts are crucial regulators of angiogenesis during skeletal development and bone regeneration [155,156]. However, in adult mice, vascular density can be further increased upon stimulation of βARs in osteoblasts via β2AR-dependent induction of VEGF synthesis in osteoblasts [157]. This increase in VEGF and bone vascular density induced by βAR stimulation was shown to promote breast cancer cell bone metastasis, as VEGF blockade or β2AR deficiency in osteoblasts specifically reduced both vascularity and the number of metastatic bone lesions formed after intracardiac injection of triple negative MDA-MB-231 breast cancer cells and ISO administration [157]. Activation of βARs in breast cancer cell lines and osteoblasts also induces the production of IL-6, which has the potential to stimulate the βAR-driven pro-angiogenic pathway by promoting the expression of VEGF [81,158,159,160,161] and the proliferation of breast cancer cells [162,163]. Therefore, a first mechanism of action whereby sympathetic nerves can promote skeletal breast cancer metastasis is through a VEGF-dependent neo-angiogenic switch upon β2AR signaling in osteoblasts and stimulation of bone marrow vascular density, which increases the likelihood of circulating breast cancer cells colonizing the skeleton. 

The arrest of circulating cancer cells into distant organs plays a central role in the process of tumor metastasis. In this context, the adhesive interaction between tumor cells and the vascular endothelium is important. Extravasation, which brings circulating tumor cells from the blood stream into the distant tissue stroma, is initialized by attachment and rolling of tumor cells along the endothelium via selectins, such as E- and P-selectin [164,165,166,167]. This process is followed by cancer cell arrest and tight adhesion mediated by integrins, before cancer cells transmigrate through the vascular endothelium [164,167]. This transient interaction between cancer cells and endothelial cells is promoted by inflammatory cytokines such as IL-1 and TNFα, whose action is to increase the expression of adhesion proteins on the endothelium [168,169,170,171,172]. IL-1β was shown to promote the adhesion of cancer cells to endothelial cells in vitro, and administration of IL-1β to mice increased the number of lung and skeletal metastases, while reducing IL-1β activity in melanoma mouse models reduced tumor burden and metastases [173,174,175,176]. In vitro experiments demonstrated that βAR stimulation of osteoblasts induced their expression of IL-1β [158]. Moreover, the conditioned medium from ISO-treated osteoblasts increased the expression of E- and P-selectin in endothelial cells and promoted the adhesion of human MDA-MB-231 breast cancer cells to these endothelial cultures, while preincubation with an IL-1β neutralizing antibody significantly reduced this effect [158]. Although the in vivo relevance of these data has not yet been experimentally addressed, a second level of SNS action on the bone microenvironment in the context of skeletal metastasis could thus be a stimulatory effect on osteoblast-derived IL1β, leading to enhanced adhesive properties of the bone vascular network for metastatic cancer cells and promotion of the arrest of these cells into the skeleton, in a E/P-selectin-dependent manner.

Whether metastatic cancer cells survive and grow in their new environment after extravasation depends on multiple cell-intrinsic and stromal-derived factors. Subjecting mice to chronic immobilization stress (CIS, known to transiently activate sympathetic nerves and the HPA axis) prior to injection of cancer cells increased the efficiency of skeletal metastasis compared to non-stressed control mice [141]. These findings indicated that sympathetic activation creates a pro-metastatic microenvironment that promotes the establishment of circulating cancer cells into the skeleton. This is in line with the aforementioned pro-angiogenic and pro-adhesive effect of β2AR signaling. The administration of the β-blocker propranolol reduced this effect, confirming the contribution of endogenous sympathetic activation and βAR signaling to this mechanism [141]. Further experiments demonstrated that the homing/migration properties of receptor activator of nuclear factor kappa-Β ligand (RANKL) were involved in this context. RANKL is expressed by osteoblasts and osteocytes as well as T cells and chondrocytes [177,178,179,180,181], and its expression in osteoblasts and osteocytes is readily increased upon βAR stimulation [141,145]. Besides its osteoclastogenic properties, RANKL is known to have a pro-migratory activity for RANK-expressing cancer cells [84,114,182,183,184,185]. High RANK expression (the receptor for RANKL) in the tumors of breast cancer patients was associated with poor prognosis and the combined expression of RANK and CXCR4 in breast cancer patients predicted recurrence of bone metastases [186,187]. In vitro, the observation that the conditioned medium of ISO-treated osteoblasts stimulated the migration of breast cancer cells, and that the RANKL decoy receptor osteoprotegerin (OPG) could block it, supported a stimulatory effect of βAR activation on breast cancer cell migration, via the upregulation of RANKL expression by the host stromal compartment [141]. These results were reinforced by the reduction in MDA-MD-231 cancer cell migration in vitro and bone metastasis in vivo following βAR stimulation and knockdown of RANK in breast cancer cells [141]. In conclusion and collectively, these results suggest that stimulation of βAR signaling in osteoblasts following activation of sympathetic nerves favors the colonization of the bone marrow environment by circulating breast cancer cells at multiple steps of the metastatic process, via (a) the promotion of a VEGF-dependent neoangiogenic switch leading to higher bone vascular density and (b) an IL-1β-dependent increase in the adhesive properties of this newly created vasculature, both of which potentially increase the likelihood of circulating breast cancer cells to arrest in bone tissues, and (c) a RANKL-dependent increase in cancer cell homing or retention into bone tissues, due to higher migratory activity toward RANKL-secreting niches (Figure 1). It is worth noting that osteocytes represent an essential source of RANKL during bone remodeling [5,188], and that β-adrenergic signals not only promote RANKL expression in osteoblasts but also in osteocytes [145]. Whether osteoblasts or osteocytes, or both, are targets of sympathetic-nerve-derived NE for its action on bone metastasis remains unknown.

Although pro-angiogenic and cancer cell bone-homing properties of βAR stimulation in the osteoblast lineage are mainly relevant to the early establishment of circulating breast cancer cells in the bone marrow environment, RANKL and IL-6 are, before all, two well-known potent osteoclastogenic cytokines [158,189,190,191]. Their high expression following βAR2 stimulation is likely to contribute to the development of osteolytic lesions and tumor progression through enhanced osteoclastogenesis once tumor cells are established in the skeleton [83,192]. Direct βAR-mediated stimulation of osteoclast differentiation has also been observed [193] and was reported to enhance the bone resorbing activity of βAR-expressing osteoclasts in vitro [194]. In addition, bone-derived growth factors, such as TGFβ, are released from the bone matrix during osteoclast-mediated resorption to promote the proliferation and survival of tumor cells [195,196,197,198,199,200,201,202]. Evidence that TGFβ plays a decisive role in tumor growth was substantiated by the demonstration that bone metastases can be effectively reduced by the TGFβ signaling blockade [200,202,203,204,205,206,207]. Therefore, stimulation of βAR in the osteoblast lineage and its effect on osteoclasts may not only favor the establishment of bone metastases at early disease stage but also fuel the feed-forward cycle of bone destruction at later stages of the bone metastatic process, by promoting osteoclastogenesis and the release of growth factors from the bone matrix [208]. The more recent discovery of the deleterious impact of bone-metastasis-derived TGFβ on the muscular system [209,210,211] also implies that high sympathetic nervous system activity would be associated with muscle weakness in the context of skeletal metastasis. In addition to their bone resorption function, osteoclasts are, as their progenitors, antigen-presenting cells [212]. Depending on their origin and environment, these cells induce regulatory CD4^+^ and CD8^+^ T cells [213,214,215] or inflammatory TNFα-producing CD4^+^ T cells [213]. Recently, an immunosuppressive subset of Cx3cr1^+^ osteoclasts was identified as promoting a pro-metastatic bone marrow microenvironment through its strong immunosuppressive capacity and upregulation of immunosuppressive checkpoint molecules including programmed death ligand 1 (PD-1L), herpes virus entry mediator (HVEM), and galectin 9 [216]. The immunosuppressive capacity of osteoclasts has also been described in the context of multiple myeloma, where the production of immunosuppressive molecules such as indoleamine 2,3-dioxygenase (IDO) and PD-L1 by osteoclasts protects myeloma cells against T-cell-mediated cytotoxicity [217]. Whether direct βAR activation in osteoclasts affects their immune function and whether this could entail an immunosuppressive and pro-metastatic microenvironment in the bone marrow remains to be investigated. Lastly, osteoclasts were shown to secrete miRNAs that trigger cancer cell proliferation, tumor cell survival, and angiogenesis [218,219,220]. Whether stimulation of adrenergic signals enhances this release and thus promotes a pro-tumorigenic microenvironment has yet to be addressed.

## 4. Treatment Strategies to Limit Metastatic Cancer Cell Engraftment into the Skeleton

Despite the successful use of anti-resorptive molecules such as bisphosphonates to limit bone pain and fracture as standard-of-care for patients with established bone metastases [221,222,223], the development of drug resistance and the persistence of minimal residual disease hampers cancer remission. An important implication of the findings summarized above is that βAR blockade should reduce the levels of VEGF, IL-1β, and RANKL produced by osteoblasts (and presumably osteocytes) in response to sympathetic activation, and thus offer some level of protection against bone metastatic events. It is important to emphasize that, in addition to a number of social stressors, the diagnosis of cancer and its treatment have a significant impact on the stress level of patients [44,224,225,226,227,228]. β-blockers are used for the treatment of multiple conditions, including congestive heart failure, hypertension, migraines, infantile hemangioma, and thyrotoxicosis [229]. The safety profile of these drugs is well-described, it is not associated with an increased incidence of breast cancer [230,231], and several clinical and preclinical studies support their use as adjuvant therapy in the treatment of breast cancer [45,46,47,48,49,95,96,97,98,99,100,101,102,232,233,234]. This beneficial effect of β-blockers on breast cancer patient survival, reduction of tumor recurrence, and reduced incidence of metastasis (summarized by Conceição and colleges [58]), remains, however, controversial because most of these studies present limitations, including retrospective design, small population size, difficulty in assessing duration of β-blocker treatment, or lack of data about comorbidities and intake of other medications. Nevertheless, improved relapse-free survival [45] and reduced metastasis and tumor recurrence were observed in breast cancer patients receiving β-blockers [46,48], and consistent with what was observed in some preclinical studies [141], a positive correlation was observed in patients taking propranolol the year before breast cancer diagnosis, specifically regarding tumor invasion and metastatic involvement at diagnosis, as well as reduced cancer-specific mortality [47]. This observation and the fact that there was no association between post-diagnostic use of β-blockers and breast cancer-specific mortality and progression [103] support the notion that metastatic dissemination, and not metastatic tumor growth, is the most relevant target of sympathetic nerves and that β-blocker therapy could improve prognosis. Of note, synergy was detected between β-blockers and chemotherapy in mice in terms of anti-tumor effect at the primary site and improved survival [235], thus further supporting the use of β-blockers concurrently with chemotherapy in the management of breast cancer. Although blockade of βAR signaling represents an attractive strategy to complement current therapeutic options, preclinical and adequately powered clinical trials focusing on overall survival and cancer recurrence are still needed to assess the therapeutic potential of β-blockers to restrain breast cancer bone metastasis. Whether the protective effect of non-selective β-blockers on skeletal metastasis is mediated by action on β1AR or β2AR also remains uncertain. It should be noted that β1AR blockade but not non-selective βAR blockade reduced bone resorption and had a favorable effect on cancellous BMD in clinical studies [94]. This action of β-blockers on BMD is another incentive to further evaluate this class of drugs for the treatment of cancer patients at risk of bone loss and fracture, especially since there are concerns regarding rare side-effects of current treatments targeting osteoclasts, such as osteonecrosis of the jaw or atypical femoral fractures [236]. One should also be aware of a recent study showing that the effect of β-blockers on overall survival may vary depending on the different subtypes of breast cancer—in contrast to improved relapse-free survival in patients with triple negative breast cancer [45,48,233,237], the pre-existing use of β-blockers in patients with advanced-stage HER2-positive breast cancer prior to anti-HER2 therapy initiation was associated with reduced overall survival compared to patients without β-blocker intake [238].

## 5. Conclusions

Preclinical studies in mice demonstrated that stimulation of β2AR in osteoblasts promotes bone vascular density and the colonization of the bone microenvironment by metastatic breast cancer cells, and that pharmacological inhibition of βAR signaling by β-blockers inhibits skeletal colonization by metastatic cancer cells. These observations support the contribution of sympathetic nerves to the process of breast cancer bone metastasis, but there are still many unanswered questions to fully understand how these nerves change the behavior of metastatic cancer cells and the properties of their stroma along the multiple steps of the metastatic process. Variables in this equation include the variety of adrenergic receptors expressed by tumor cells and stromal cells, differential signaling upon various ligands, and the multiple homeostatic mechanisms regulating endogenous sympathetic NE release and AR signaling in both presynaptic and post-synaptic cells, as well as the contribution of various immune cells. Much remains to be done to broaden findings in different animal models of bone metastasis, in immunocompetent animals, and with different types of cancer cells. Clinical studies are also still needed to confirm the beneficial effect of low cost, safe β-blockers on cancer cell bone metastasis and relapse-free survival.

## Figures and Tables

**Figure 1 cancers-13-02887-f001:**
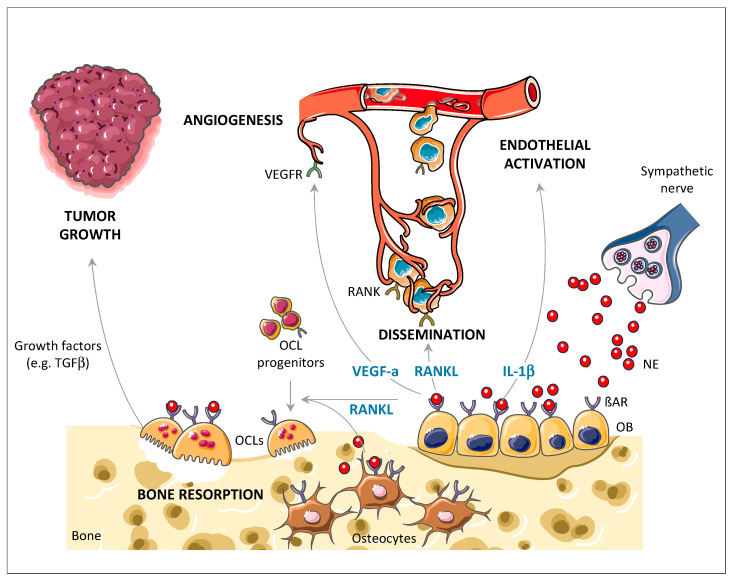
Overview of the different mechanisms by which sympathetic nerve activation promotes the establishment of metastatic breast cancer cells in the skeleton. Secretion of NE from sympathetic nerve terminals in the bone microenvironment stimulates post-synaptic βARs on various target cells, including bone and cancer cells. βAR activation on osteoblasts results in increased IL-1β, VEGF-a, and RANKL secretion, leading to endothelial activation and neoangiogenesis as well as tumor stromal retention and increased osteoclastogenesis, respectively. It also promotes tumor growth through increased osteoclastogenesis, bone resorption, and subsequent release of bone-derived growth factors such as TGFβ. βAR, β-adrenergic receptor; OB, osteoblast; OCL, osteoclast; NE, norepinephrine; RANK, receptor activator of nuclear factor kappa-Β; RANKL, receptor activator of nuclear factor kappa-Β ligand; VEGF-a, vascular endothelial growth factor A; VEGFR, vascular endothelial growth factor receptor.

## Data Availability

No new data were created or analyzed in this study. Data sharing is not applicable to this article.

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
