# Peer review of "Mechanisms Supporting the Use of Beta-Blockers for the Management of Breast Cancer Bone Metastasis"

_cancers, 2021, doi:10.3390/cancers13122887_

Round 1

Reviewer 1 Report

This manuscript needs revision

Abstract needs to be more concise and clear

In section 1) elaborate  more on  metastatic  process and cancer angiogenesis also  elaborate  more on clinics of breast cancer and role of beta-adregergic receptors as well as chemotherapy  use with  beta AR blocker on effect on metastasis including overall survival of the patient  

Elaborate  more  on triple negative breast cancer and  estrogen, progesterone and HER2 and correlation to beta AR

Figure 1 needs more clarification

Authors need to show connection between beta-AR and breast cancer  metastasis  and cancer angiogenesis

Author Response

We thank the reviewers for their effort, time and constructive comments. We have addressed them in the text (major changes in blue) and below are brief responses to each of them. We hope the review is now suitable for publication.

Best regards

F. Elefteriou, PhD

Responses:

Abstract needs to be more concise and clear:

The abstract is already in the shortest form allowed by this Journal and cannot be shortened. We have modified it slightly to improve clarity.

In section 1) elaborate  more on  metastatic  process and cancer angiogenesis also  elaborate  more on clinics of breast cancer and role of beta-adregergic receptors as well as chemotherapy  use with  beta AR blocker on effect on metastasis including overall survival of the patient  

We have elaborated further about the metastatic process (p2), cancer angiogenesis (p5) and association between b-blockers and chemotherapy (p4, 8).

Elaborate  more  on triple negative breast cancer and  estrogen, progesterone and HER2 and correlation to beta AR

We have mentioned the variability of a/bAR expression across cell lines and their molecular subtypes and added a reference about a recent review where these correlations where provided.

Figure 1 needs more clarification. Authors need to show connection between beta-AR and breast cancer  metastasis  and cancer angiogenesis

We have added some clarifications in the figure legend to summarize mechanisms and link between bAR stimulation, neoangiogenesis in the metastatic site and tumor colonization of this site (p6). We hope it is now clearer.

Reviewer 2 Report

The authors review the literature surrounding the influence beta blocker administration has on breast cancer metastasis to bone. The authors cover describe the role of the sympathetic nerve system in the skeleton and review evidence from the literature of stimulation in bone metastasis. The authors finalise their review with an overview of treatment strategies to limit metastatic cancer cell growth within the bone environment.

A recent review by Conceicao et al (2021) discussed the role of the SNS in breast cancer and metastasis and included a section on beta-blockers. The authors should include a reference to this paper published in Bone Research in their manuscript and highlight the additional information they provide that is not covered in this previously published review.

Author Response

We thank the reviewers for their effort, time and constructive comments. We have addressed them in the text (major changes in blue) and below are brief responses to each of them. We hope the review is now suitable for publication.

Best regards

F. Elefteriou, PhD

Responses:

We thank the reviewer for bringing this very recent review to our attention. Conceicao et al have indeed done an excellent job at reviewing this research area. We have added reference to some of their very useful synthetic tables (p2/4/8), and added in the introduction that this review focused more on the early stages of metastasis, compared to this manuscript that mainly deals with metastatic colonization of the skeleton. Despite some level of necessary overlap, these two reviews are thus complementary.

Reviewer 3 Report

Overall, this is a clear, concise, and well-written review article. The authors have drafted the introduction really well providing sufficient background information. They have provided a sufficient information to highlight the role of beta-blockers as a therapeutic option in managing bone metastasis observed in breast cancer patients. The authors have described current knowledge and perspectives about the mechanisms underlying use of beta-blockers. The manuscript will have a broad appeal not only to breast cancer researchers but also to researchers studying bone metastasis associated with other cancer types.

Author Response

We thank the reviewers for their effort, time and constructive comments. We have addressed them in the text (major changes in blue) and below are brief responses to each of them. We hope the review is now suitable for publication.

Best regards

F. Elefteriou, PhD

Responses:

We thank the reviewer for his/her positive comments.

Round 2

Reviewer 1 Report

Authors did not  corrected manuscript as recommended

Authors did not correct  “Elaborate  more  on triple negative breast cancer and  estrogen, progesterone and HER2 and correlation to beta AR”

Author Response

We have developed the paragraph on a/bAR expression and molecular subtype (in red, p4). We hope it addresses the reviewer's comment.